# COMMIT0: LIBRARY GENERATION FROM SCRATCH

**Wenting Zhao**[1]    **Nan Jiang**[1]    **Celine Lee**[1]    **Justin T Chiu**[2]
**Claire Cardie**[1]    **Matthias Gallé**[2]    **Alexander M Rush**[1]

[1]Cornell University    [2]Cohere

wz346@cornell.edu

## ABSTRACT

With the goal of benchmarking generative systems beyond expert software development ability, we introduce COMMIT0, a benchmark that challenges AI agents to write libraries from scratch. Agents are provided with a specification document outlining the library's API as well as a suite of interactive unit tests, with the goal of producing an implementation of this API accordingly. The implementation is validated through running these unit tests. As a benchmark, COMMIT0 is designed to move beyond static one-shot code generation towards agents that must process long-form natural language specifications, adapt to multi-stage feedback, and generate code with complex dependencies. COMMIT0 also offers an interactive environment where models receive static analysis and execution feedback on the code they generate. Our experiments demonstrate that while current agents can pass some unit tests, none can yet fully reproduce full libraries. Results also show that interactive feedback is quite useful for models to generate code that passes more unit tests, validating the benchmarks that facilitate its use. We publicly release the benchmark[1], the interactive environment[2], and the leaderboard[3].

## 1    INTRODUCTION

AI agents have been increasing rapidly in ability, particularly in domains such as problem-solving, math, and coding. Tasks related to software development have been particularly promising areas due to both their clarity of evaluation and economic value. This has motivated the release of several coding benchmarks in recent years (Hendrycks et al., 2021a; Chen et al., 2021; Zhuo et al., 2024). A notable example is SWE-bench (Jimenez et al., 2024), which assesses the ability of agents to generate patches to resolve real-world GitHub issues. While critical, these tasks generally remain within the skill set of an experienced software engineer. If LLM systems continue to improve at current rates, these tasks will be completely solvable.

We are interested in benchmarks that exist further beyond both the frontier of expert human ability as well as current model ability. Specifically, tasks that experts struggle to solve but can still be fully specified and reliably verified. Software engineering is an appealing domain for this, as the process of developing actual implementations of functions is very complex. Nevertheless, humans can fully specify the desired behavior of functions and validate them through unit testing.

With this goal in mind, we introduce COMMIT0, a benchmark that tests an agent's ability to generate a software library from scratch. This task is especially challenging – large, real-world libraries are notoriously difficult to design, often requiring hundreds of engineers and years of development. Nonetheless, this task remains verifiable without requiring humans to solve it directly. Humans can provide specifications that outline the library's API and write unit tests to verify whether the API has been implemented correctly.

COMMIT0 extends beyond existing benchmarks in several ways. Central to COMMIT0 is *interactive feedback*. Due to the complexity of generating a library, it is improbable, or likely impossible, that an agent could generate a complete, working version in one shot. Instead, the benchmark is constructed

---

[1]https://huggingface.co/datasets/commit0/commit0
[2]https://github.com/commit-0/commit0
[3]https://commit-0.github.io/

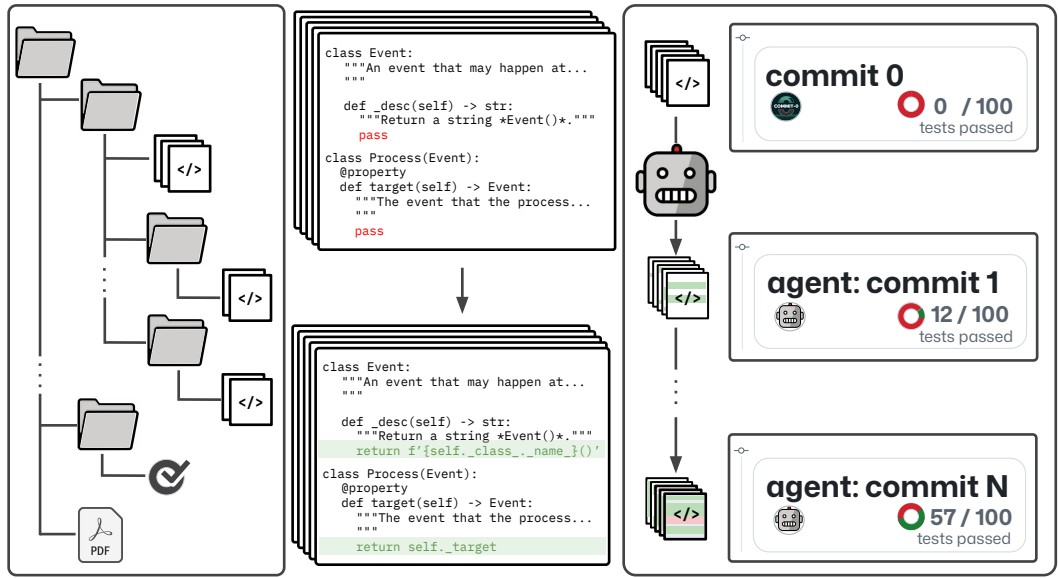

Figure 1: An overview of COMMIT0. Given a starter repository with empty function body, a specification, and a suite of unit tests, agents are required to produce an implementation of the library that passes all unit tests.

such that must adapt to multi-stage feedback such as unit test errors. Libraries also feature *complex dependencies*. Implementing one function in a library involves calling other functions, and therefore Agents need to identify the right order to implement the functions. Finally COMMIT0 features *long-context processing*: agents must navigate specifications of hundreds of pages, and generate thousands of functions, both of which require processing texts in a long context.

While our main focus is the benchmark itself, we also introduce a prototype agent SDE-I for completing the benchmark. The agent introduces a basic method for traversing the complex library dependencies, uses best-in-class LLMs to process long contexts, and responds to the interactive feedback of the system. To perform code completion SDE-I uses a state-of-the-art coding agent.

We empirically evaluate this system on COMMIT0. Our experiments show that with a state-of-the-art LLM without feedback, it can pass 17% unit tests in the easier libraries but can only pass 6% in all libraries. We find that iterating on error messages from unit tests improves the pass rate of unit tests to 26% on the easier libraries, demonstrating the utility of leveraging execution feedback. Finally, conditioning on relevant files – i.e., ensuring the agent considers related file dependencies and context – further enhances performance.

## 2 RELATED WORK

**Evaluation of LMs.** Recent benchmarks for evaluating agents focus on knowledge-intense, exam-style questions in domains ranging from grade-school mathematics to quantum mechanics (Hendrycks et al., 2021c; Srivastava et al., 2023; Hendrycks et al., 2021b; Rein et al., 2023). While these questions are challenging, with some requiring PhD level knowledge, they are often short and easy to memorize, and they only require few steps of sequential reasoning. In contrast, Commit0 requires reasoning over a long horizon. Mastering the ability to develop full repositories requires considering many files, many unit tests, and complex static analysis feedback.

**Software engineering benchmarks** Existing benchmarks for code generation focus on specific aspects of the software engineering pipeline. Program synthesis benchmarks evaluate code generation, for example, HumanEval (Chen et al., 2021), MBPP (Austin et al., 2021), BigCodeBench (Zhuo et al., 2024), CodeBenchGen (Xie et al., 2024), and Classeval (Du et al., 2023). Segmented benchmarks, such as DevBench (Li et al., 2024), separately evaluate different aspects such as code design, code generation, and unit test synthesis. R2E (Jain et al., 2024) introduces a more chal-

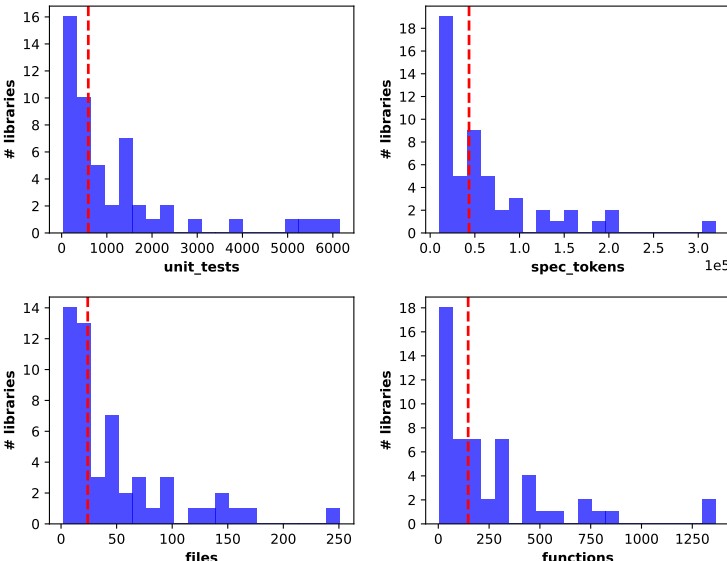

Figure 2: Basic statistics of COMMIT0. The red dotted line denotes the median. **Top left**: the distribution of the number of unit tests in a library; **top right**: the distribution of the number of tokens in a specification; **bottom left**: the distribution of the number of source files in a library; **bottom right**: the distribution of the number of public functions to be implemented in a library.

lenging task by requiring function generation that involves dependencies within and across files. SWE-bench (Jimenez et al., 2024) provides a more holistic evaluation of a model's ability to resolve pull requests, requiring the incorporation of repository-level context. However, the amount of context necessary to resolve a specific pull request varies greatly and is small on average. These previous benchmarks focus on generating one or a few functions and are thus manageable via static one-shot code generation. In contrast, COMMIT0 requires generating an entire codebase consisting of numerous interdependent functions, which necessitates a series of refinements based on execution results to pass all unit tests.

**Software engineering agents**   Recent work has made impressive progress in developing software engineering agents that operate on repositories (Yang et al., 2024; Zhang et al., 2023; Wang et al., 2024a). Commit0 proposes a software agent that not only operates at the repository-level but also self-corrects given test feedback. Our method extends prior work on self-correction (Madaan et al., 2023; Shinn et al., 2023) to the larger-scale problem of repository generation.

## 3   THE COMMIT0 BENCHMARK

COMMIT0 benchmarks an agent's ability to generate a functioning library from scratch. It consists of 54 Python libraries covering a wide range of topics, including machine learning, networking, databases, and data visualization. Given, (1) a specification document that contains both texts and images, (2) a starter repository with both unit tests and files to fill in, agents are tasked with completing the implementation of the API described in the specification document. Libraries are *prepared* by removing the core source code from their repo in a systematized manner.

The agent is provided with a specification in PDF format and a starter repository containing a source code directory and a test directory. The task is to make edits to the repository. In practice, the model generates modified versions of the source code files. We then replace the original files with the modified ones and perform a commit to save the repository state. The model is allowed to run unit tests interactively along with code generation. For evaluation, we clone this commit in a clear repository and run unit tests. The model's performance is measured only by the pass rate of these unit tests.

To prevent the model from copying source code, we restrict access to original GitHub repositories via web retrieval. However, the model is allowed to use the web for general knowledge lookup. For

instance, if it needs to implement a radar plot for visualization, it is allowed to search for relevant information online.

Figure 2 presents basic statistics of COMMIT0. The top left of Figure 2 shows the distribution of unit tests across all libraries. Approximately 30 libraries have fewer than 1,000 unit tests, but the distribution shows a long tail. The top right illustrates the distribution of tokens in the specifications, with even the smallest specification containing over 10,000 tokens. Most libraries have fewer than 50,000 tokens, though the longest specification reaches up to 300,000 tokens. The bottom left displays the distribution of files, where the majority of libraries have fewer than 50 files, but some exceed 100. Finally, the bottom right shows the number of public functions to be implemented. While most libraries have fewer than 250 functions, this number can exceed 1,300 in some cases.

## 3.1 LIBRARY SELECTION

We focus on Python libraries due to their widespread use, abundant data resources, and strong ecosystem support. To select a set of high-quality libraries for models to implement, we design three sets of filtering criteria.

**Library requirements.** We restrict libraries to be Python-only. Specifically, the library needs to contain over 95% Python code. The library also needs to have native Python implementations instead of using Python wrapping libraries in other languages. Finally, the library must support testing with *pytest*.

**Specification requirements.** We identify libraries that have comprehensive specifications. The specification must have its own webpage rather than a plain README page. The specification document must cover both a user guide which describes how the library is intended to be used and a comprehensive API reference that defines the input and the output of a function. The specification should both describe in natural language what are the inputs and outputs and specify the types of inputs and outputs.

**Unit test requirements.** We include libraries with comprehensive unit tests to test the implementation of a library, while having understandable tests that are feasible to run in an interactive system. We limit the libraries to those with over 90% of code that can be covered by unit tests. We filter libraries whose unit tests take over than 30 minutes to run on a single CPU and the libraries where a significant number of unit tests can only be run on GPU.

To compile the list of libraries included in COMMIT0, we consider both generally popular Python libraries and PyPI packages with top download counts[4]. We follow the annotation guideline[5] to filter out the libraries that satisfy the criteria described above.

We create two dataset splits: *lite*, which includes libraries with fewer functions to implement, and *all*, which contains all libraries. Lite has a total of 16 libraries. Due to the complexity of COMMIT0 and budget constraints, we focus most of our evaluation on COMMIT0 lite.

## 3.2 BENCHMARK CONSTRUCTION

**Ensuring Replicability.** A key aspect of COMMIT0 is replicable running of unit tests across all the libraries, which depends on the correct setup of development environments. To achieve this, we annotate setup commands for each library. We begin by annotating a specific commit of the library repository, which is used to extract installation requirements and generate the starter repository. The installation requirements typically include a compatible Python version, necessary `pip` packages, and an installation command. Some libraries may also require system-level dependencies, such as `clang`. Finally, we annotate the pytest command, the directory containing the unit tests, and the source code directory.

**Preparing Libraries.** We prepare a library for COMMIT0 by removing its core code in a systematic way. We assume that a library contains public functions which are accessible to users, and private functions which are not supposed to be called. This is often not enforced explicitly by Python

---

[4]`https://hugovk.github.io/top-pypi-packages/`

[5]We include the annotation guideline in Appendix.

but is upheld by convention. To determine if a function is a public function, we check if it has an associated docstring. To prepare COMMIT0, we replace the function body of all public functions to be empty (pass) and remove all private functions entirely. We perform these code modifications by first parsing each Python file into an abstract syntax tree, performing transforms on the syntax tree, and converting back to source code.[6]

**Preparing Specifications.** Specifications exist in different forms. Some libraries have pure text descriptions while others have extensive figures to demonstrate how the libraries work. For example, seaborn is a data visualization library; it uses figures to demonstrate the expected outcomes of API functions. To unify the format, we convert all specifications to the PDF format. Specifically, starting from the main documentation page, we crawl the webpage as well as all the internal links recursively and save them as a PDF.

## 3.3 INTERACTIVE ENVIRONMENT

A key feature of COMMIT0 is its interactivity. Generating an entire library in a single attempt is challenging for an agent; it may need to iteratively incorporate feedback to refine the implementations. COMMIT0 provides an interactive environment that allows agents multiple sources of feedback, including unit testing, static analysis, and coverage analysis.

**Unit Test Feedback** Unit tests are crucial for validating the specified behavior of functions. The results from unit tests provide valuable information about implementation issues, including error types and execution traces. Our interactive environment allows agents to execute an arbitrary number of unit tests for any library in parallel. The primary challenge in this process is the need to set up environments for each library to run the unit tests. To address this, we create a Docker image for each library and execute the tests in these isolated environments. This setup allows agents to simultaneously develop and run unit tests across all libraries using the pre-built images.

**Static Analysis Feedback** Our interactive environment also offers comprehensive static analysis feedback, including linting and type checking, as an additional corrective signal. We apply a standardized linter and configuration file across all libraries to ensure consistency. Specifically, we use ruff as our linter[7]. For reproducibility, we release both the Docker images and the linter configuration file alongside our benchmark.

**Coverage Feedback** Coverage analysis serves as another valuable signal. For instance, if a unit test passes in one run but fails in another, the difference in coverage can help identify which lines of code are causing errors by comparing the differences in coverage. To provide this coverage information, we use pytest-cov and leverage the pre-built isolated environments described above to run the coverage analysis in a reproducible way.

## 4 THE SDE-I AGENT

Challenges involving complex multi-file dependencies and interactive feedback are challenging for current agentic systems (Yang et al., 2024). To test the difficulty level of COMMIT0, we design a prototype interactive agent. SDE-I performs a basic software engineering loop. It writes functions, runs unit tests, and iteratively edits the code based on error messages. The agent operates in three stages as shown in Figure 3.

**Stage 1: Draft initial implementations.** SDE-I drafts an initial implementation for each function. The first challenge is determining the appropriate unit for generation. Implementing all functions at once is impractical, as concatenating them often exceeds the model's maximum context length. However, implementing each function in isolation loses the broader context of the other functions. To balance this, we choose to treat all functions within a single module[8] as one generation unit.

The second challenge is managing complex dependencies between modules since implementing a module often requires understanding which other modules it depends on. SDE-I performs a

---

[6]done with the ast library

[7]github.com/astral-sh/ruff

[8]In Python, a file is a module.

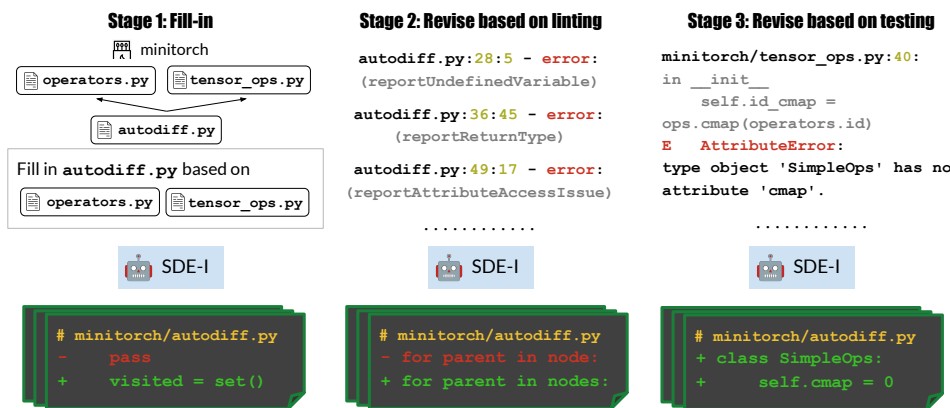

Figure 3: Overview of SDE-I.

topological sort on imports of source code modules. It constructs a directed acyclic graph (DAG) of the library, where each node represents a module. If a module imports others, the imported modules are set as its parent nodes. In the case of conditional import cycles, a random edge is removed to break the cycle. SDE-I then proceeds by filling in the modules in the order determined by the topological sort. When implementing a module, it also includes the content of all modules that the current module imports. In this step, SDE-I entirely ignores whether the generated code is executable or not.

**Stage 2: Refine based on static analysis.** SDE-I improves the initial implementations by running static analysis to detect and correct issues related to code style, syntax, and type errors. This static analysis helps enforce coding standards and catch potential problems before running more resource-intensive tests. SDE-I appends these results to the context and generates revised versions of where issues were located. This process is repeated until all errors are resolved or the maximum number of runs is reached.

**Stage 3: Refine based on unit test results.** SDE-I refines the implementation further by running unit tests to ensure functional correctness. A challenge similar to the first stage arises: running all unit tests simultaneously may generate error messages that exceed the model's maximum context length. However, unit tests are naturally grouped by functionality, with tests for related features typically organized within the same test module. We leverage this structure by executing unit tests one module at a time. The results are then incorporated into the context, allowing SDE-I to revise the code based on the error messages. This process is iterative, with SDE-I continually revising the code until all tests pass or a predefined limit on test runs is reached.

**Implementation** The SDE-I is implemented to be modular for the underlying coding system and language model. For code generation, we default to the Aider framework.[9] Aider's interface allows us to define a prompt, a lint command, and a test command. We construct a message that includes a prompt to fill in the missing function body, along with texts from specifications and any necessary import modules. For the LLM, we evaluate several model families known for their strong performance on coding benchmarks. Specifically, we consider GPT-4o-mini (Hurst et al., 2024), o1-preview (OpenAI, 2024), Claude 3.5 Sonnet[10], DeepSeek-V2.5 (Guo et al., 2024), Llama-3.1-8B-Instruct, Llama-3.1-70B-Instruct, Llama-3.1-405B-Instruct (Dubey et al., 2024), and Codestral[11].

## 5 RESULTS

To assess the effectiveness of each stage in the SDE-I agent, we evaluate ablated versions of the method where we apply a fixed number of stages. We summarize the results on COMMIT0 lite in Table 1. (Note that we skip stage 2 for OpenAI o1-preview due to its high costs.) Among the three models, Claude 3.5 Sonnet consistently delivers the best performance across all three stages.

---

[9]aider.chat
[10]anthropic.com/news/claude-3-5-onnet
[11]mistral.ai/news/codestral/

|  | Stage 1 | Stage 2 | Stage 3 |
|---|---|---|---|
| OpenAI o1-preview | $17.34_{105.92}$ | - | $21.46_{913.35}$ |
| Claude 3.5 Sonnet | $17.80_{1.55}$ | $18.79_{12.47}$ | $29.30_{99.39}$ |
| DeepSeek-V2.5 | $16.55_{1.43}$ | $11.61_{10.21}$ | $25.43_{26.41}$ |
| Llama-3.1-8B-Instruct | $6.03_{1.47}$ | $0.23_{1.78}$ | $0.37_{2.77}$ |
| Llama-3.1-70B-Instruct | $7.10_{10.85}$ | $1.83_{11.25}$ | $2.49_{24.82}$ |
| Llama-3.1-405B-Instruct | $8.08_{7.94}$ | $1.76_{12.20}$ | $4.95_{29.10}$ |
| Codestral | $6.34_{0.30}$ | $6.34_{0.36}$ | $7.41_{1.99}$ |

Table 1: Unit test pass rates across three stages of SDE-I on COMMIT0 lite. Subscripts are corresponding costs in US dollars.

| Library | Total | Stage 1 | Stage 2 | Stage 3 |
|---|---|---|---|---|
| babel | 5663 | 0 | 0 | 0 |
| cachetools | 215 | 173 | 179 | 179 |
| chardet | 376 | 3 | 25 | 3 |
| cookiecutter | 367 | 108 | 102 | 16 |
| deprecated | 171 | 73 | 80 | 151 |
| imapclient | 267 | 0 | 0 | 31 |
| jinja | 851 | 0 | 0 | 0 |
| marshmallow | 1229 | 456 | 338 | 456 |
| minitorch | 230 | 0 | 0 | 0 |
| parsel | 206 | 10 | 10 | 0 |
| portalocker | 36 | 15 | 1 | 15 |
| pyjwt | 259 | 11 | 11 | 128 |
| simpy | 140 | 20 | 17 | 94 |
| tinydb | 201 | 27 | 38 | 64 |
| voluptuous | 149 | 0 | 0 | 0 |
| wcwidth | 38 | 6 | 6 | 1 |

Table 2: Pass rate on COMMIT0 lite across three stages of SDE-I.

Surprisingly progressing from Stage 1 to Stage 2 results in a decline in performance with the open-weights models. As discussed in the qualitative analysis in Section 6, although the static analysis feedback provides useful guidance for fixing bugs, the model struggles to apply it effectively, often introducing additional errors. This issue particularly affects the less capable models. However, moving from Stage 2 to Stage 3 consistently improves the average pass rate, demonstrating the value of utilizing unit test feedback. In the cost-constrained setting, Codestral in stage 1 has the best performance (6.34%) under $1, and Claude 3.5 Sonnet in stage 3 has the best performance (%29.3) under $100.

Table 2 shows the pass rate for each library using Claude 3.5 Sonnet at each stage. At the individual library level, the results are mixed. In many cases, when models attempt to address static analysis issues and unit test feedback, they inadvertently introduce new errors. However, for libraries like *deprecated*, *parsel*, and *tinydb*, Claude 3.5 Sonnet shows the greatest improvement from execution feedback.

|  | Stage 1 | Stage 2 | Stage 3 |
|---|---|---|---|
| GPT-4o-mini | $2.87_{8.73}$ | $1.42_{27.22}$ | $4.24_{123.74}$ |
| Claude 3.5 Sonnet | $6.12_{20.46}$ | - | - |
| DeepSeek-V2.5 | $2.33_{14.94}$ | $2.95_{22.58}$ | $4.93_{84.11}$ |

Table 3: Unit test pass rates at the first stage of SDE-I on COMMIT0 all. Subscripts are corresponding costs in US dollars.

**Results on COMMIT0 all.** We summarize the results in Table 3. Due to the high API cost of Claude 3.5 Sonnet, we only run it for the first stage of the SDE-I agent on COMMIT0 all. We note that running just the first stage with Claude 3.5 Sonnet already outperforms running all stages with

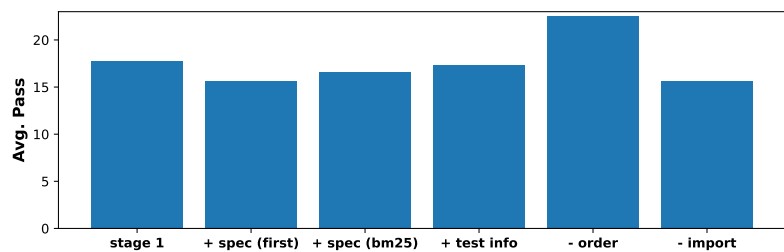

Figure 4: Ablations on different configurations of what context is provided to SDE-I.

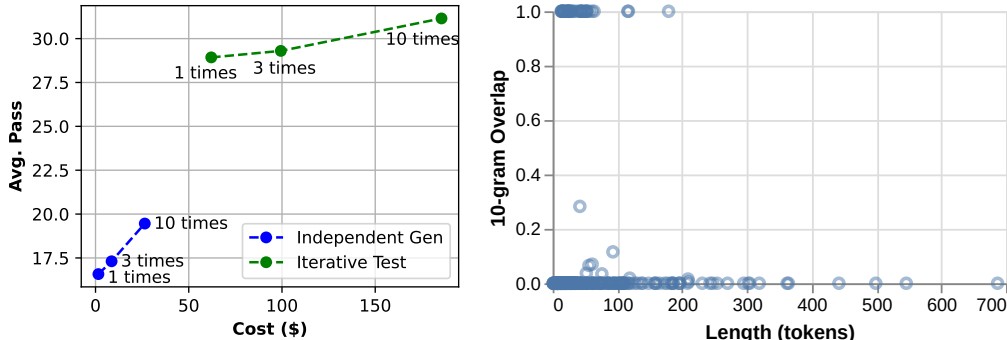

Figure 5: **Left**: Cost-constrained test-time approaches. The blue line shows test-time scaling in stage 1, where SDE-I generates $n$ copies of a module and picks the copy that passes the most unit tests. The green line shows test-time scaling in stage 3, where SDE-I iterates on unit test feedback for $n$ iterations. **Right**: 10-gram overlaps between Claude-generated functions and reference functions. Each dot is a function. We sort the 10-gram overlap by function lengths.

GPT-4o-mini or DeepSeek-V2.5. In the compute-constrained setting, GPT-4o-mini in stage 1 has the best performance (2.87%) under \$1, and Claude 3.5 Sonnet in stage 1 has the best performance (%6.12) under \$100.

**Results on state-of-the-art software development agent: OPENHANDS.** We test OPENHANDS (Wang et al., 2024b), the best-performing agent on SWE-bench. To have OPENHANDS take full advantage of the specifics of COMMIT0, we share the list of unit test IDs, static analysis commands, and test commands with the agent. With this deep integration, OPENHANDS passes 42.95% of unit tests on COMMIT0 lite (a 13.65% improvement compared to SDE-I) and 15.25% on COMMIT0 all (a 9% improvement). To understand this improvement, we analyze the *TinyDB* example, where OPENHANDS improved from 64 passed tests to 174. We attribute the success of OPENHANDS to its better debugging capabilities. Unlike SDE-I, which often repetitively generates the same fix for a bug, OPENHANDS is able to explore different solutions.

## 6 ANALYSIS

**Ablations.** We conduct ablation studies based on the first stage of SDE-I in Figure 4. First, we investigate whether including information from the specifications and tests can help LLM agents pass more unit tests. Since the length of the specification often exceeds the maximum context length of the LMs, we feed only the first 10,000 tokens from the specification. For test, we append the prompt to include test modules. Surprisingly, both additions reduce performance. We hypothesize that much of the specification and tests are irrelevant to implementing specific modules, which may distract the model. To better leverage the specification, the agent will likely need to first filter out only the relevant information. To verify this hypothesis, we perform retrieval to obtain 10,000 tokens. Specifically, we break the specification into chunks of 1,000 tokens and retrieve the top 10 chunks to include in the context. With the same number of tokens, using BM25-retrieved tokens yields a higher unit test pass rate, suggesting that agents can benefit from more relevant context.

Next, we explore whether filling in functions following the order found by topological sort is helpful. Interestingly, we find that a random-ordering approach leads to more passed unit tests (22% compared to 17%). Upon further inspection, we found conditioning on incorrect implementations of previous modules was the issue. Relying on incorrect implementations appears to be more harmful than relying on empty ones. Lastly, we assess the impact of excluding relevant modules from the context. As expected, excluding these relevant imports results in fewer passed unit tests.

Table 4: Qualitative example for using static analysis feedback to revise function implementations.

**Test Results Before**

```
PASSED tests/test_utils.py::test_work_in
PASSED tests/test_utils.py::test_work_in_without_path
```

**Lint Feedback**

```
cookiecutter/utils.py:38:5: ANN201 Missing
return type annotation for public function 'work_in'
 37 | @contextlib.contextmanager
 38 | def work_in(dirname=None):
    |     ^^^^^^^ ANN201
 39 |     """Context manager version of os.chdir.
    = help: Add return type annotation
```

**Revised Implementation**

```
 @contextlib.contextmanager
- def work_in(dirname=None):
+ def work_in(
+     dirname: Optional[Union[str, "os.PathLike[str]"]] = None
+     ) -> Any:
     """Context manager version of os.chdir.
@@ -44,11 +50,12 @@ def work_in(dirname=None):
     try:
         if dirname is not None:
             os.chdir(dirname)
-         yield
+         yield None
     finally:
         os.chdir(curdir)
```

**Test Results After**

```
Failed tests/test_utils.py::test_work_in
    - TypeError: 'NoneType' object is not an iterator
Failed tests/test_utils.py::test_work_in_without_path
    - TypeError: 'NoneType' object is not an iterator
```

**Test-time Scaling.** An interesting question is how unit test pass rates scale with more test-time compute. We address this question with two experiments. First, we sample a module 1, 3, and 10 times, picking the best implementation based on pass rates before proceeding to the next module. Additionally, we test whether continuous iterations on unit test feedback will eventually enable agents to pass all unit tests. We conducted an experiment where we applied unit test feedback over different numbers of iterations: 1, 3, and 10. We summarize the results on the left of Figure 5. We observed that, in both cases, unit test pass rates improve with more test-time compute.

**Library Memorization.** COMMIT0 tests models on existing libraries, versions of which are may be part of their training data. This raises the possibility that LLMs may be simply recalling these libraries from memory, but not reasoning about the specifications and unit tests. (We not that a similar question exists for many other coding benchmarks.) To explore whether current models are primarily memorizing the libraries, we calculate the 10-gram overlap between the generated and reference libraries. We create a mapping from function IDs to their corresponding function bodies for public functions. We then compute the 10-gram overlap between the generated implementation and the reference implementation for each function. The overlaps are sorted by function length. The results, presented on the right of Figure 5, indicate that LMs either fully memorize the functions or produce implementations that are significantly different. Shorter functions are more frequently

memorized. Given the low total test accuracy, the observed degree of memorization is not a major factor for the benchmark.

Table 5: Qualitative example for using unit test feedback to revise function implementations.

**Test Results Before**

```
FAILED tests/test_condition.py::test_ior_with_or_cond
FAILED tests/test_condition.py::test_ior_with_and_cond
```

**Unit Test Feedback**

```
>        event._value = event._callback(event)
E        AttributeError: 'Initialize' object has no attribute '_callback'.
         Did you mean: 'callbacks'?
```

**Revised Implementation**

```
-        event._ok = True
-        event._value = event._callback(event)
-        event._processed = True
+        try:
+            if hasattr(event, '_callback'):
+                event._value = event._callback(event)
+                event._ok = True
+            elif hasattr(event, 'callbacks'):
+                for callback in event.callbacks:
+                    callback(event)
+                event._ok = True
+            elif isinstance(event, Process):
+                if event._target is None:
+                    raise RuntimeError('Invalid yield value "None"')
+                event._resume(event._target)
+            else:
+                event._ok = True
+        except Exception as e:
+            event._ok = False
+            event._value = e
+            if not getattr(event, '_defused', False):
+                raise
+        finally:
+            event._processed = True
```

**Test Results After**

```
PASSED tests/test_condition.py::test_ior_with_or_cond
PASSED tests/test_condition.py::test_ior_with_and_cond
```

**Qualitative Analysis.** In the first example, we show how static analysis feedback might hurt performance. Presented in Table 4, the function implemented in stage 1 successfully passes the unit tests. However, the agent misinterpreted the static analysis feedback, which requested type annotations. In response, the agent adds `Any` as the return type and modifies the function to include `yield None` to match the type. Unfortunately, since a generator cannot return NoneType, this introduces a new error. In the second example, we show test feedback can improve performance. Presented in Table 5, the unit test feedback points out that the attribute '_callback' is missing. The agent thus revised the implementation by adding an if statement to check for relevant attributes, hence passing the unit tests.

## 7 CONCLUSION

We introduce COMMIT0, a challenging task that requires LMs to generate libraries from scratch. Our task provides signals through unit test feedback, including both error messages and execution traces. It also offers comprehensive static analysis feedback including type checking. This task is meant to be beyond the level of most human experts, and currently seems beyond what state-of-the-art LLMs are capable of. We hope that it can serve as both as progress benchmark for AI development as well as spurring new agent architectures and methods.

## LIMITATIONS

One limitation of COMMIT0 is that we exclusively include Python libraries in our evaluation, which may limit the generalizability of our findings to other programming languages. Additionally, our experimental environment requires a well-specified document and a full suite of unit tests for each library, which may not reflect the typical development process where documentation and tests are often incomplete or evolving. While these assumptions may not be entirely realistic in every real-world scenario, they are closely aligned with the principles of test-driven development (TDD), where rigorous testing and clear documentation are integral parts of the development process.

Furthermore, our reliance on unit tests as the primary means of evaluating the correctness of the generated code introduces additional limitations. Unit tests may not cover all possible use cases or edge cases, potentially allowing erroneous or suboptimal code to pass unnoticed. This overreliance on testing may mask underlying issues that would be evident in a thorough code review or through integration testing. As a result, the agent's ability to generate code that passes all tests does not necessarily equate to producing robust, high-quality software.

Another concern is the risk associated with having agents generate code that humans may find difficult to read or maintain. Automatically generated code might be syntactically correct and functionally adequate to pass tests, but it may lack clarity, proper documentation, or adherence to coding standards and best practices. This can lead to maintainability issues, as future developers may struggle to understand or modify the codebase, increasing the risk of introducing bugs or vulnerabilities in subsequent updates. The opaqueness of machine-generated code also raises questions about accountability and traceability in software development.

## ETHICS STATEMENT

COMMIT0 consists of forked public repositories whose licenses permit our use. Our study does not involve human participants, and does not rely on human task workers for data collection. We do not gather any data, including personal data, from GitHub.

Additionally, code generation without human oversight may raise potential ethical issues. For example, agents could introduce security vulnerabilities or violate licensing terms. Ensuring that the generated code is safe, compliant, and ethically sound may require human intervention, which our current benchmark does not account for.

Finally, automating software engineering is a challenge that has both potential benefits and harms. Our release of COMMIT0 serves to measure progress towards this challenge.

## REPRODUCIBILITY

We release the COMMIT0 benchmark in its entirety, along with all methods and results. We also provide the code for reproducing the dataset, so that it may be used for synthesizing data.

## ACKNOWLEDGEMENTS

We thank Ofir Press, Xingyao Wang, Graham Neubig, Tanya Goyal, and Sanjiban Choudhury for the helpful discussions. We thank Modal for providing us with credits to run unit tests using their service. We thank annotators from Cohere for providing annotations for additional libraries. AMR, CL, and NJ are supported by NSF CAREER-2037519, NSF-2242302, and NSF Grant DRL-2229873.

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

## A DATA ANNOTATION

Table A lists what we annotated for each library in order to execute the unit tests.

| Annotation | Description |
|---|---|
| Repository Name | Owner and repository name |
| Commit or Tag | Annotate either a specific commit or a version tag (recommended). |
| Python Version | Python version that is compatible with the specified code state |
| Packages | Path to 'requirements.txt' that contains packages to be installed |
| Pip packages | Additional pip packages to install |
| Install | Installation command (must be in editable mode and include test dependencies) |
| Pre-install | System-level dependencies (e.g., `apt-get`, `clang`, etc.) |
| Specification | URL link to the project specification, preferably a PDF link |
| Test Command | `pytest` command for running unit tests |
| Test Directory | Directory where unit tests are located |
| Source Directory | Directory where the source code is located (e.g., `web3/` for `web3.py` library) |

Table 6: Annotations for setting up executing unit tests

## B IMPLEMENTATION DETAILS

```
Here is your task:
You need to complete the implementations for all functions (i.e.,
those with pass statements) and pass the unit tests.  Do not change
the names of existing functions or classes, as they may be referenced
from other code like unit tests, etc.
IMPORTANT: When you generate code, you must maintain the original
formatting of the function stubs (such as whitespaces), otherwise we
will not be able to search/replace blocks for code modifications, and
therefore you will receive a score of 0 for your generated code.
```

Figure 6: The prompt provided to SDE-I at stage 1.

**Prompt.** We present the stage-1 SDE-I prompt in Figure 6.

## C LIBRARY ANNOTATION GUIDELINES

We share the library annotation guidelines[12] that we provide to annotators. To incorporate new libraries into COMMIT0, one can follow these guidelines to produce the necessary annotations for the libraries.

## D ABLATIONS

| | Fill in by Module | Fill in by Pass Statement |
|---|---|---|
| Avg. pass | 17.80 | 12.42 |

Table 7: Comparison of average pass rates between filling in by module and by function.

---

[12]`https://tinyurl.com/commit0-annotation-guidelines`

**Generation units.** Here we analyze the generation unit for filling in libraries from scratch. We compare filling in by module and by pass statement and present the results in Table 7. We find that filling in by pass statement leads to significantly worse results than filling in at the module level. When generating code for one pass statement, models tend to focus solely on local contexts and often overlook interactions with other parts in the modele. In contrast, generating code at the module level enables models to develop a holistic understanding of a file. For example, in the following `portalocker.py` file, when filling in for the only pass statement, the agent terminates without any modification.

```
import os
import typing
from . import constants, exceptions
LockFlags = constants.LockFlags

class HasFileno(typing.Protocol):
    pass
LOCKER: typing.Optional[typing.Callable[
[typing.Union[int, HasFileno], int], typing.Any]] = None
if os.name == 'nt':
    import msvcrt
    import pywintypes
    import win32con
    import win32file
    import winerror
    __overlapped = pywintypes.OVERLAPPED()
elif os.name == 'posix':
    import errno
    import fcntl
    LOCKER = fcntl.flock
else:
    raise RuntimeError('PortaLocker only defined for nt and posix platforms')
}
```

As a result, all the unit tests failed in the collection stage:

```
    ImportError while loading conftest '/testbed/portalocker_tests/conftest.py'.
portalocker_tests/conftest.py:8: in <module>
    from portalocker import utils
portalocker/__init__.py:37: in <module>
    lock = portalocker.lock
E   AttributeError: module 'portalocker.portalocker' has no attribute 'lock'
```

However, when the agent is asked to fill in the whole module, it returns the following message:

```
Thank you for providing the task details and the content of the relevant files.
I'll focus on completing the implementations for the functions with `pass`
statements in the `portalocker/portalocker.py` file. Since there are no explicit
`pass` statements in the provided code, I'll assume that we need to implement
the locking and unlocking functionality for both Windows (nt) and POSIX systems.

Let's start by implementing the `lock` and `unlock` functions for both platforms.
```

Following this message, the agent produces the code:

```
if os.name == 'nt':
    [...]
    def lock(file, flags):
        if flags & constants.LockFlags.EXCLUSIVE:
            lock_type = win32con.LOCKFILE_EXCLUSIVE_LOCK
```

```
        else:
            lock_type = 0
        if flags & constants.LockFlags.NON_BLOCKING:
            lock_type |= win32con.LOCKFILE_FAIL_IMMEDIATELY
        hfile = msvcrt.get_osfhandle(file.fileno())
        try:
            win32file.LockFileEx(hfile, lock_type, 0, -0x10000, __overlapped)
        except pywintypes.error as exc_value:
            if exc_value.winerror == winerror.ERROR_LOCK_VIOLATION:
                raise exceptions.LockException(fh=file)
            else:
                raise
    def unlock(file):
        hfile = msvcrt.get_osfhandle(file.fileno())
        try:
            win32file.UnlockFileEx(hfile, 0, -0x10000, __overlapped)
        except pywintypes.error as exc_value:
            if exc_value.winerror == winerror.ERROR_NOT_LOCKED:
                # File was not locked.
                pass
            else:
                raise
elif os.name == 'posix':
    [...]
    def lock(file, flags):
        locking_flags = fcntl.LOCK_EX
            if flags & constants.LockFlags.EXCLUSIVE
            else fcntl.LOCK_SH
        if flags & constants.LockFlags.NON_BLOCKING:
            locking_flags |= fcntl.LOCK_NB
        try:
            fcntl.flock(file.fileno(), locking_flags)
        except IOError as exc_value:
            if exc_value.errno == errno.EACCES or exc_value.errno == errno.EAGAIN:
                raise exceptions.LockException(fh=file)
            else:
                raise
    def unlock(file):
        fcntl.flock(file.fileno(), fcntl.LOCK_UN)
else:
    raise RuntimeError('PortaLocker only defined for nt and posix platforms')
```

