# OpenReview forum: "Commit0: Library Generation from Scratch"
_ICLR.cc/2025/Conference — ICLR 2025 Poster_

### Official Review · Reviewer_xSyA · 2024-10-27

**Soundness:** 3
**Presentation:** 3
**Contribution:** 3
**Rating:** 8
**Confidence:** 4

**Summary:**

Summary: The Commit0 paper proposes a new benchmark for the coding proficiency of large language models. Simple existing coding benchmarks are typically done on smaller well defined problems with a simpler interaction model of providing a problem description to the LLM and getting back the LLM’s generated code solution. Commit0 tries to create problems to solve with the complexity a human software engineer would have to solve in a real job, like implementing a whole set of related functions from the spec for the library. Commit0 also includes the iterative nature of software development where the LLM’s code solutions are run through lint checks, and test case checks and the test results and LLM generated code are passed back to the LLM to allow it to iterate and attempt to fix any issues found.

Commit0 also includes an example LLM agent SDE-I to implement the functionality needed to solve the benchmark proposed. The SDE-I agent calls leading LLM models from places like Anthropic and OpenAI to provide baseline scores for the Commit-0 benchmark. The SDE-I agent defines the basic dataflow, prompts and interfaces needed to complete the Commit0 benchmark tasks.

The goal of this benchmark is to more accurately assess how well LLMs can do solving coding problems that are a lot more like what real human SDE have to do at work. Improvements in the benchmark will come from better LLMs being developed, and improvements to SDE-I to create better prompts and work-flows to pass to the LLMs the best context in terms of related code and task descriptions of the functionality required in the context window passed to the LLM.

**Strengths:**

Commit0 is a much more realistic measure of how well an LLM will do software development on a task that matches what real human software developers have to do in their jobs.

Including the iterative nature of software development by passing back lint and test errors to the LLM to allow it to attempt further edits and improvements to the code is great.

Having a whole set of functions to implement to support a scenario is great to force the LLM to handle the complexity of tasks real human software engineers have to deal with.

**Weaknesses:**

W0: The computational burden to solve the full Commit0 test set is immense, so the cost to run this benchmark might be prohibitive for many labs to participate. What does it cost currently?

W1: Real developers have to produce specs for the high-level design and shared data structures, and also unit tests and docs for all the private functions they are writing. That sort of thing I guess could be all done inside the SDE-I agent, but the ability for the agent to ask itself, what is the spec for this internal function, what are the unit tests for to verify this internal function is correctly implemented and allowing the agent to iterate on the errors it finds by running it’s own unit tests on it’s own internal (and public) functions and being able to ask itself if the function it wrote completely implements all aspects of the spec in a “reflexion” sort of way I think will be important to enable surpassing human levels of coding in the SDE-I agent.

W2: With training LLMs there are these scaling laws about if you use a bigger model, or train for longer on more data – the results just get better.  Similarly for the SDE-I agent if it’s allowed to independently generate each function multiple times and then pick the best implementation in a “reflexional” sort of manner, or allowed to iterate more times to fix errors, and to restart from scratch if it is still failing a function’s tests after N calls – the error rate can be pushed lower with more CPU/GPU power applied. So comparing different SDE-I agents – it seems like there has to be budget of how many tokens-in/tokens-out are allowed to compare different approaches. If you are running your LLM local, then maybe it’s wall clock time.  If you are calling models in the cloud, maybe it’s total cost in $$$, but it just feels like to find the best SDE-I design the total compute between different approaches has to be held constant, or the total cost of calling models in the cloud has to be constant, for fair comparisons between SDE-I agents and the LLMs they use.

**Questions:**

Q0: What does it cost to run the Commit0 tests to get the numbers presented in the paper? It would be nice to see more baselines like from Llama 8,70,405 or Mistral or the leading coding LLMs, even if just for the lite/small testset.

Q1: I think defining a smaller subset of the benchmark was smart in the paper, given the cost to solve the whole thing. Would it make sense or be useful to take current simple benchmarks (humaneval, etc) and plug them in as tiny benchmark sets in Commit0, so the power of being able to iterate on the lint and test results can be checked out by smaller labs. Also to debug and tune future SDE-I agents, it might be nice to have proof of concept small Commit0 test sets to test against.  I suggest repackaging common coding datasets in Commit0 format because some effort has been made to exclude (humaneval, etc) from LLM training datasets, so as a fast easy Commit0 dataset they could serve some use.

Q2: Would a benchmark for Commit0 leaderboard of best results for $20 compute, $200 compute make sense?  Like should I use OpenAI’s most expensive model for fewer calls/iterations or use OpenAI’s smaller less expensive models for more calls/iterations? Basically level the playing field so it’s the best agent for a fixed cost – use any LLM you want – but you only have $20 or $200 to work with. Design a test subset and pick a dollar $$$ amount that is reasonable for labs to participate.  In the SDE-I agent basically you run through all the problems in the benchmark in 1 pass and then you just keep iterating on your results till you run out of money.

---

> ### Author Response · Authors · 2024-11-19
>
> We thank the reviewer for the thoughtful review!
>
> W0: Solving commit0 is too costly.
> ---
>   * This is a valid concern. For this reason, we make the lite split of the benchmark.
>   * We expect the costs of running LLMs to continue decreasing significantly. As evidence: in 2022, the leading model, text-davinci-003, was 100 times more expensive to run than GPT-4o mini in 2024, despite GPT-4o mini being a much stronger model.
>   * The following table shows how much it costs to run SDE-I on lite and all.
>
> |                   | Stage 1 | Stage 2 | Stage 3 |
> |-------------------|---------|---------|---------|
> | OpenAI o1-preview | $105.92 | -       | $913.35 |
> | Claude 3.5 Sonnet | $1.55   | $12.47  | $99.39  |
> | DeepSeek-V2.5     | $1.43   | $10.21  | $26.41  |
>
> Table 1: costs to run SDE-I on commit0 lite.
>
> |                   | Stage 1 | Stage 2 | Stage 3 |
> |-------------------|---------|---------|---------|
> | GPT-4o-mini       | $8.73   | $27.22  | $123.74  |
> | Claude 3.5 Sonnet | $20.46  | -       | -       |
> | DeepSeek-V2.5     | $14.94  | $22.58   | $84.11      |
>
> Table 2: costs to run SDE-I on commit0 all.
>
> W1: Commit0 doesn’t consider producing specifications and unit tests.
> ---
>   * We note that there are indeed many scenarios in software development where a spec exists and the coder has to implement it, such as implementing a new programming, database backend, or a web browser. This is often referred to as test-driven development (TDD). Therefore, an agent that performs well on commit0 demonstrates its utility in TDD contexts.
>   * We also agree that a current limitation of Commit0 is that the specification and unit tests are given. This is a design choice we explicitly made in order to ensure that evaluation is unambiguous, repeatable, and robust. Note that our benchmark is still very difficult, requiring code generation at a much larger scale than previous benchmarks.
>   * In future work, we will investigate relaxing the assumption that specifications and tests are fully observed. Robust evaluation given incomplete specifications is an open problem and we are very excited about it. Indeed, methods like Reflexion -- where models iteratively improve their outputs through self-assessment -- are valuable for enabling agents to generate and refine their own specifications and tests. These methods can help models not only generate code but also reason about its correctness, completeness, and alignment with the intended functionality.
>
> W2: Fair comparisons of baselines require setting fixed budgets / need more extensive study on test-time scaling approaches for commit0
> ---
>   * Thank you for highlighting the importance of controlling computational resources when comparing different SDE-I agents. We agree that to ensure fair comparisons, it's crucial to account for the amount of compute associated with each approach.
>   * While measuring compute by calculating the exact number of FLOPs would be ideal, it's not feasible due to the proprietary nature of some models. We thus address this by approximating compute usage through the dollar cost incurred during model evaluations. We will plot the average pass rate against the total cost for each approach on our leaderboard. This allows us to visualize the trade-offs between performance and computational expense, effectively identifying methods that lie on the Pareto frontier. By doing so, we can highlight approaches that offer the best performance for a given cost.
>   * Thank you for suggesting test-time scaling approaches for commit0. We note that researchers are also exploring varying levels of test-time computation for SWE-Bench. Considering the difficulty of solving commit0, it would be exciting to see advancements using these test-time scaling methods. We encourage researchers to explore these strategies, and on our leaderboard, we will recognize approaches where unit test pass rates effectively scale with increased test-time computation.
>   * As the reviewer suggested, we plot (1) independently generating each module 1, 3, 10 times and picking the best implementation based on pass rates, and (2) iterating 1, 3, 10 times to fix errors. We plot these six runs in here: https://drive.google.com/file/d/1xjXgreigXSzmL5F7EZePf4MWjIJKWYPT/view?usp=sharing
>
> |                            | 1 time | 3 times | 10 times |
> |----------------------------|--------|---------|----------|
> | Avg. pass | 16.58  | 17.31   | 19.46    |
> | cost                       | $1.55  | $8.62   | $26.42   |
>
> Table 3: Best of $n$ independent generation.
>
> |                            | 1 time | 3 times | 10 times |
> |----------------------------|--------|---------|----------|
> | Avg. pass | 28.93  | 29.30   | 31.15    |
> | cost                       | $62.07 | $99.39  | $185.54  |
>
> Table 4: Iterative refinement based on test feedback.

---

> > ### Author Response · Authors · 2024-11-19
> >
> > Q0: Adding results on Llama 8,70,405 or Mistral on commit0 lite.
> > ---
> >   * Thanks for the great suggestion. We have added these results below for the suggested models.
> >
> > |                | Stage 1 | Stage 2 | Stage 3 |
> > |----------------|---------|---------|---------|
> > | Llama-3.1-8B   | 6.03%   | 0.23%   | 0.37%   |
> > | Llama-3.1-70B  | 7.10%   | 1.83%   | 2.49%   |
> > | Llama-3.1-405B | 8.08%   | 1.76%   | 4.95%   |
> > | Codestral      | 6.34%   | 6.34%   | 7.41%   |
> >
> > Table 5: Results on open-source code models on commit0 lite.
> >
> > Q1: Adding HumanEval as part of commit0 harness.
> > ---
> >   * This is a great suggestion. In the next commit0 release (in December), we will include both HumanEval (easy) and SWE-bench (medium) in our hardness. Our interactive environment will support running tests for both of these benchmarks.
> >
> > Q2: Adding fixed budgets for baseline approaches.
> > ---
> >   * This is a great suggestion. We plot the results by budgets: https://drive.google.com/file/d/1wQMjpleC3_k812ilf1-7Z_c4mmiWnu9c/view?usp=sharing
> >   * For commit0 lite, we will set the budgets to be 50 USD and 100 USD. We will pose this cost constraint to future submissions. Currently, under the 100 USD budget, SDE-I agent (stage 3) with Claude 3.5 Sonnet achieves the best performance. Under the 50 USD budget, SDE-I agent (stage 3) with DeepSeek achieves the best performance.

---

> ### Comment · Reviewer_xSyA · 2024-12-03
>
> I read the updated paper and the comments from the authors answering the many questions. I think the paper has improved and have increased my score accordingly, I hope to see it accepted to the conference. I think this benchmark sets up a framework to push the LLM community forward in solving more difficult and realistic coding tasks and provides example baselines for folks to easily build on with their own ideas.

---

### Official Review · Reviewer_ffwo · 2024-11-02

**Soundness:** 2
**Presentation:** 3
**Contribution:** 3
**Rating:** 6
**Confidence:** 4

**Summary:**

This paper introduces a new benchmark commit0, evaluating agents abilities of writing python libraries from scratch. In this benchmark, agents are provided with starter repos, specifications, and unit tests to start with. Then, the agents' final commits with the implementation of the libraries would be tested by the unit tests. They also designed a prototype agent SDE-I that operates in three stages.

**Strengths:**

- The authors curated a new benchmark commit0 that could assess agents' abilities of implementing various python functions. This benchmark could help with evaluating and thus further improving agents' code and repo generation abilities.
- The authors introduced a new framework SDE-I which could assist the repo generation process for agents.
- The authors perform some ablation studies that might provide insights into how agents utilize additional information of specifications/tests

**Weaknesses:**

- The authors did not provide comprehensive evaluation of their proposed agent SDE-I on the full benchmark commit0 that they curated. They only provide full results of SDE-I of stage 1 on the full benchmark. Thus, their results might not fully reflect the actual abilities of SDE-I on commit0. It would be good to have the results of all stages of SDE-I on commit0.
- The authors did not evaluate other existing agents on commit0, which makes it less clear how current agents perform on such tasks. It would be interesting to have results of agents based on frameworks like ReACT or CodeAct on commit0.
- The authors do not provide example instances from the commit0 benchmark, nor example trajectories by SDE-I on commit 0. It would also be nice to have code for synthesizing new data and example benchmark evaluation code. With example instances provided, it will be more clear what are tasks like in commit0. It would be good to have 1 or 2 examples in the main text of the paper and include more in the appendix.
- The authors did not provide the average cost of SDE-I on commit0. For example, it would be good to have a table containing the average cost of the proprietary models.
- The authors did not make clear how they prevent agents from accessing existing online implementations of the python functions via web browsing. It would be good to have a more detailed explanation maybe in the appendix.
- When performing ablation study, the authors found a decrease in performance of the agent when additionally provided with the specifications or tests, and hypothesized that this might be due to the specifications/tests not relevant to implementation. However, the authors only feed the first 10000 tokens of the full specification / test to the agent. The prompts being incomplete might affect the performance. It would be more interesting to see if including relevant details of the specification / test would help improve agent performances.

**Questions:**

- It would be nice if the authors could provide further explanation on how they prevent agents from accessing existing online implementations of the python functions via web browsing.
- It would be nice if they authors could provide more example instances from commit0 and provide example trajectories by SDE-I.

---

> ### Author Response · Authors · 2024-11-19
>
> We thank the reviewer for the thoughtful review!
>
> W0: Limited results on the all split of commit0.
> ---
>   * Obtaining results for all stages of SDE-I using current state-of-the-art models is prohibitively expensive. Running o1-preview for the first and the third stages on the lite split alone cost $913, and we are limited by our budget. However, to do the best we can to test out all stages of SDE-I on the all split, we test Deepseek V2.5 and GPT-4o-mini.
>
> |             | Stage 1 | Stage 2 | Stage 3 |
> |-------------|---------|---------|---------|
> | deepseek    | 2.33%   | 2.95%   |  4.93%|
> | GPT-4o-mini | 2.87%   | 1.42%   | 4.24%   |
>
> Table 1: Results on commit0 all split using DeepSeek V2.5 and GPT-4o-mini.
>
> W1: Limited evaluation on other agents.
> ---
>   * We would like to note that none of the existing agents can automatically work for library generation without modification.
>   * To include an evaluation of another agent, we selected the top agent on the [SWE-bench leaderboard](https://www.swebench.com/) as of today. The leading agent is OpenHands + CodeAct v2.1 (`claude-3.5-sonnet-20241022`), released on October 25th. We collaborated with the OpenHands team to ensure the best performance of their agent. The results are provided below.
>
> |             | SDE-I | OpenHands |
> |-------------|---------|---------|
> | Lite    | 29.30   |  43.20  |
> | All | 6.12   | 15.62   |
>
> Table 2: New results on OpenHands.
>
> W2 & Q1: Missing commit0 examples and trajectories.
> ---
>   * Commit0 example: https://anonymous.4open.science/r/minitorch-3E5F/README.md. A specific processed module can be seen here: https://anonymous.4open.science/r/minitorch-3E5F/minitorch/autodiff.py
>   * Example trajectory on all stages of tinyDB with Sonnet: stage 1 (https://drive.google.com/file/d/1DVpK3yVXvG0zSnMFzOIB-B6wvhC47BJd/view?usp=sharing), stage 2 (https://drive.google.com/file/d/15G56gA5xLnSwC3O7PRuJDwVP_Niq1gJ0/view?usp=sharing), stage 3 (https://drive.google.com/file/d/1AgKcz8_YGIFy3zeoSX3J3HepicKTIx5P/view?usp=sharing)
>   * We will update the appendix to include links to __all__ examples and __all__ trajectories.
>
> W3: Missing costs of SDE-I agents.
> ---
>   * Thanks for the great suggestion. We will add the following table to the paper.
>
> |                   | Stage 1 | Stage 2 | Stage 3 |
> |-------------------|---------|---------|---------|
> | OpenAI o1-preview | $105.92 | -       | $913.35 |
> | Claude 3.5 Sonnet | $1.55   | $12.47  | $99.39  |
> | DeepSeek-V2.5     | $1.43   | $10.21  | $26.41  |
>
> Table 3: Costs for running the SDE-I agent across different stages.
>
> W4 & Q0: Missing description for how to prevent agents from accessing source code.
> ---
>   * We automatically verify whether an agent has retrieved information from GitHub by detecting the presence of 'github.com' or 'raw.githubusercontent.com' in the retrieval URLs. If these URLs are detected, the corresponding generation will be voided. Thus far, the agent hasn’t retrieved code from GitHub. We will clarify this in the paper.
>
> W5: Feeding the first 10k tokens of spec docs to agents might be sub-optimal.
> ---
>   * Thanks for the great suggestion. We have added a retrieval experiment where we first tokenize the specification, and break it into chunks of 1000 tokens. When generating a module, we retrieve the top 10 relevant chunks and include them as part of the context. Using BM 25 to retrieve the most relevant 10k tokens does lead to an improvement.
>
> |                         | Avg pass |
> |-------------------------|----------|
> | Spec (first 10k tokens) | 15.67    |
> | Spec (BM25 10k tokens)  | 16.62    |
>
> Table 4: Feeding specification documents with a retrieval component.

---

> ### Comment · Reviewer_ffwo · 2024-11-24
>
> Thank you for your response.
>
> > We would like to note that none of the existing agents can automatically work for library generation without modification.
>
> This is not true, as can be shown by the high results of OpenHands on Commit0 Lite.
>
> > We selected the top agent on the SWE-bench leaderboard as of today.
>
> It seems that OpenHands has higher performance on both Commit0 Lite and All compared to SDE-I. I have several questions:
>
> - What's the cost of OpenHands on Commit0.
> - Given that SDE-I is specifically tuned for library generation, what could be the reason for OpenHands achieving much higher performance than SDE-I.
>
> Your responses to W0, W2, W3, W4, and W5 addressed my concerns. However, it would be good to submit a revised version of your paper for the rebuttal, with the promised additions.

---

> > ### Author Response · Authors · 2024-11-24
> >
> > We thank the reviewer for engaging with us!
> >
> > * **Regarding integration efforts**: We would like to clarify that OpenHands did **not** automatically work for Commit0. To have OpenHands take full advantage of the specifics of Commit0, we provide OpenHands with our pre-built docker images. This provides the agent access to our full interactive environments. The total integration is 600+ lines of code. We are happy to see our efforts lead to new state-of-the-art results.
> > * **Costs**: Unfortunately, the cost incurred by OpenHands is not clear to us -- it doesn't track costs, as documented [here](https://github.com/All-Hands-AI/OpenHands/issues/3685).
> > * **Analysis of OpenHands improvement**: We attribute the success of OpenHands to (1) better debugging capabilities and (2) better tool use. For (1), unlike SDE-I, which often repetitively generates the same fix for a bug, OpenHands is able to explore different solutions. For (2), we observe that OpenHands frequently uses bash tools such as ``cat`` and ``grep`` that help it more effectively navigate through large codebases.
> > * **Revised PDF**: We have uploaded a new version of our paper that includes the experiments suggested by all the reviewers. To maintain the anonymity of our submission, please note that the links to the examples and trajectories are currently placeholders. We will replace the placeholders with the actual links in the public version.

---

> > > ### Comment · Reviewer_ffwo · 2024-11-26
> > >
> > > Thanks for the fast response. I have raised my scores accordingly.

---

### Official Review · Reviewer_FDu7 · 2024-11-03

**Soundness:** 3
**Presentation:** 3
**Contribution:** 2
**Rating:** 6
**Confidence:** 5

**Summary:**

This paper introduces commit0, a benchmark for evaluating programming agents on generating software libraries from scratch given comprehensive specifications and unit tests. The authors then present SDE-1, a prototype agent that demonstrates different aspects of designing agents for this task such as drafting an implementation, feedback via linters, refining with execution feedback.

**Strengths:**

1. **Ambitious benchmark even if not for current gen models.** I particularly like that the benchmark pushes beyond current codegen evaluation setups by targeting full library implementations. Although this might seem too difficult a task currently, having such a benchmark could produce interesting solutions in the space (a la swebench), esp. for such long-horizon tasks.

2. **Multiple sources of feedback.** It's nice to see the benchmark itself integrate several sources of feedback such as linting, coverage, and unit tests. This is usually moved to the agent/system being evaluated, but I think having some standard ones such as linting within the benchmark makes comparing solutions more standardized.

**Weaknesses:**

**Missing actionable insights.** While commit0 tackles a newer task compared to related work like HumanEval, SWEBench, and R2E, I'm not convinced the insights gathered are interesting or new. While the community can attempt to extract more, it would be nice for the paper to suggest a few directions even with primitive experiments. For instance, the diminishing returns on iterating with execution feedback has already been shown in prior work. I'm wondering if there's any evidence that this task will bring new insights for designing code generation systems/agents or training better models.

**SDE-1 Design** The authors mention that to balance the model's broader context of other functions, they use an entire module as one unit of generation with module-level dependency analysis. I am not convinced that this is a good choice. A module can be quite large in popular repositories and libraries (such as the ones in commit0) spanning a few 100 lines making it quite complicated to generate. My concern is that the result "random-ordering leads to more passed tests than topological sort" is a result of the unit being too large to generate, leading to more incorrect implementations.

**Questions:**

Que: In Figure4, it's unclear what the columns "stage 1" and "+ spec" suggests. It seems that the only prompt for a commit0 task is the specification of the library. However, the "stage 1" column suggests a setting where you don't pass the spec at all? What is the prompt in this case?

---

> ### Author Response · Authors · 2024-11-19
>
> We thank the reviewer for the thoughtful review!
>
> W0: Missing actionable insights: what are the new insights commit0 can bring in addition to previous benchmarks?
> ---
>   * **Actionable direction 1: Modular Code Generation Strategies.** Commit0 presents a significant increase in task complexity compared to benchmarks like HumanEval, SWEBench, and R2E. While those benchmarks focus on generating **one or a few functions** -- manageable via static one-shot code generation -- commit0 requires generating **an entire codebase** consisting of numerous interdependent functions. This complexity introduces challenges that cannot be addressed by existing methods.
>     * Challenge: Generating all functions in one shot is impractical due to context window limitations and the intricate dependencies between functions.
>     * Primitive experiments: We conducted preliminary experiments where we treated individual files as the unit of generation and generated these files in topological order based on their dependencies.
>     * Findings: Generating code at the file level allowed some unit tests to pass, indicating progress over attempting to generate everything at once. While ordering files topologically didn't yield immediate improvements, it highlights a promising direction for optimizing generation sequences.
>     * Future Direction: Research into optimal generation units (e.g., classes, modules) and order could lead to more efficient code generation strategies.
>   * **Actionable direction 2: Advancement of Long-Context Models.** An alternative approach to generating large codebases is to utilize long-context models. commit0 can thus also function as a benchmark for long-context capabilities.
>     * Challenge: The current models have an effective maximum generation length of approximately 4k to 16k tokens, which is considerably shorter than that of a real-world codebase.
>     * Future Direction: An alternative approach to generating large codebases is to utilize long-context models. commit0 can thus also function as a benchmark for long-context capabilities.
>   * **Actionable direction 3: Enhanced Error Handling Techniques.** In benchmarks like SWEBench, debugging is typically confined to minor issues in mostly correct code (given that a correct codebase already exists), allowing models to focus on errors near the top of the call stack. In contrast, commit0 starts from an empty implementation, meaning errors can originate from any level within deep call stacks. This complexity requires models to adopt more sophisticated debugging strategies.
>     * Challenge: With errors potentially existing at any level, models need to efficiently navigate and rectify issues throughout the entire codebase.
>     * Primitive experiments: We experimented with providing models with type errors from linters before presenting them with actual execution error traces.
>     * Findings: This approach improved model performance by helping them identify problematic areas early on.
>     * Future Direction: Investigating how models can better plan and prioritize when faced with long and complex error traces could lead to advancements in effective debugging.

---

> ### Author Response · Authors · 2024-11-19
>
> W1: Current generation unit might be sub-optimal.
> ---
>   * Thank you for your feedback regarding our choice of using entire modules as the unit of generation. We understand your concern that modules in popular repositories can be quite large, potentially making them complicated for models to generate and possibly affecting the results of our ordering experiments. We would like to clarify a few points:
>     * **Module Size and Context Length:** The ablation study is conducted on the lite split, where the modules and their imported dependencies are relatively short. When generating code in random order and topological order, we include content from imported modules. In both cases, generating entire modules, with their imports included in the context, does not exceed the model's context length limitations. The only difference lies in whether these included modules have their implementations filled in at that point in the generation process. This only results in minor length differences.
>     * We have also shown that including content from imported modules is helpful. See Figure 4, excluding content from imported modules results in fewer unit tests passed.
>   * **Generation unit is not the smaller the better:** We have observed that selecting the appropriate generation unit is crucial for performance. We experimented with filling in code at the function level, but this approach produced significantly worse results than filling in at the module level. When generating code by function, models focus solely on the current function and often overlook interactions with other functions. Conversely, generating code by module enables models to develop a holistic understanding of all functions within a file. We will provide a qualitative analysis in the Appendix for further explanations.
>
> |           | Fill in by module | Fill in by function |
> |-----------|-------------------|---------------------|
> | Avg. pass | 17.80             | 12.42               |
>
> **Table**: Comparison between filling in by module and filling in by function using Claude Sonnet 3.5 in stage 1.
>
> Q0: Whether spec is included in stage 1 experiment?
> ---
>   * We apologize for the confusion and will clarify this in the paper. In the stage 1 experiment, we do **not** include the spec docs; we only provide models with the module to be filled in. The prompt is
> ```
> Here is your task: You need to complete the implementations for all functions (i.e., those with pass statements) and pass the unit tests. Do not change the names of existing functions or classes, as they may be referenced from other code like unit tests, etc. When you generate code, you must maintain the original formatting of the function stubs (such as whitespaces), otherwise we will not able to search/replace blocks for code modifications, and therefore you will receive a score of 0 for your generated code.
> ```
> One important detail is that, in the module, there are function stubs and docstring descriptions (for example: https://anonymous.4open.science/r/minitorch-3E5F/minitorch/autodiff.py). Therefore, the spec docs are not necessary for understanding the spec of the current functions but can be a useful resource for obtaining a holistic understanding of the entire library.

---

> ### Comment · Reviewer_FDu7 · 2024-11-27
>
> Thank you for the detailed response.
>
> While the insights you have detailed are correct, I still think it is quite limited to (1) managing context or (2) handling errors -- which overlap with several other code benchmarks. The generation unit aspect is interesting, but the current results with module-level generation don't reveal much as of now.
>
> > We experimented with filling in code at the function level, but this approach produced significantly worse results than filling in at the module level.
>
> This is interesting but could be result of how whether or not an invariant during generation is maintained. For instance if a function `foo(a: T1, b: T2)` with argument types `T1` and `T2` was generated first, the model needs to maintain `foo` as an invariant while generating a function `bar` that calls `foo`. The right experiment would provide the model enough context of the previously generated code as generation progresses.
>
> Overall, I appreciate the detailed responses, but would like to maintain my score.

---

### Author Response · Authors · 2024-11-24
**Paper is updated with new experiments, clarifications, etc**

We thank the reviewers again for their valuable feedback, which has helped us improve our paper. We have uploaded a new version of the paper with the following changes (highlighted in blue):

1. Added clarifications for the difference between Commit0 and the previous benchmark (FDu7)
2. New results for analyzing the optimal generation units; in Appendix (FDu7)
3. Added the prompt provided to the SDE-I agent; in Appendix (FDu7)
4. New results for on Commit0 all (ffwo)
5. New results using OpenHands to solve Commit0 (ffwo)
6. Added the associated cost for each experiment run (ffwo, xSyA)
7. Added explanations for how to prevent agents from accessing Github source code (ffwo)
8. New results for analyzing the role of retrieval when providing specifications (ffwo)
9. New results for applying test-time scaling approaches on Commit0 (xSyA)
10. New results for using Llama and codestral to solve Commit0 (xSyA)
11. Added a cost metric when discussing the main results (xSyA)

---

### Meta-Review · Area_Chair_CrRZ · 2024-12-21

**Metareview:**

The paper "Commit0: Library Generation from Scratch" introduces a novel benchmark, Commit0, designed to evaluate AI agents' capabilities in generating entire software libraries from scratch, based on provided API specifications and interactive unit tests. The benchmark aims to push beyond static one-shot code generation by incorporating long-form natural language processing, multi-stage feedback, and complex code dependencies. The authors also present an example agent, SDE-I, which operates in three stages—drafting, linting feedback, and execution feedback—to demonstrate the benchmark's functionality. Experiments reveal that while current agents can pass some unit tests, none can fully implement entire libraries, highlighting the benchmark's difficulty. The authors also validate the utility of interactive feedback in improving code generation performance.

#### Contribution
1. **Novel Benchmark**: Commit0 is a pioneering benchmark for evaluating AI agents on complex library generation tasks, addressing a gap in existing code generation benchmarks that typically focus on simpler, isolated functions.
2. **Interactive Environment**: The benchmark incorporates an interactive environment with execution and linting feedback, providing a more realistic assessment of AI agents' capabilities in real-world software development scenarios.
3. **Prototype Agent (SDE-I)**: The introduction of the SDE-I agent serves as a baseline for the benchmark, demonstrating the feasibility and challenges of the task.
4. **Actionable Insights**: The paper outlines potential research directions, such as optimizing generation units, leveraging long-context models, and improving error handling techniques, which could advance the field of code generation.

#### Weaknesses
1. **Limited Evaluation**: The evaluation of the SDE-I agent is incomplete, with full results provided only for Stage 1 on the Commit0 benchmark, limiting the understanding of its full capabilities.
2. **Lack of Comparative Baselines**: The paper does not evaluate other existing agents on Commit0, making it unclear how the proposed benchmark compares to state-of-the-art agents in related tasks.
3. **Insufficient Examples**: The absence of example instances from the Commit0 benchmark and trajectories from the SDE-I agent reduces transparency and makes it harder to understand the task specifics and agent behavior.
4. **Cost Considerations**: The computational and financial costs of running the benchmark, particularly on the full dataset, are not adequately addressed, potentially limiting its accessibility to researchers.
5. **Potential Data Contamination**: The method to prevent agents from accessing existing online implementations (e.g., from GitHub) is not clearly explained, raising concerns about fairness and validity of the benchmark results.
6. **Limited Insights**: While the benchmark is ambitious, the paper does not provide sufficiently novel insights compared to existing benchmarks, particularly regarding the design of the SDE-I agent and its generation strategy.

**Additional Comments On Reviewer Discussion:**

1. **Limited Evaluation (ffwo):**
   - **Concern**: The SDE-I agent was only evaluated for Stage 1 on the full benchmark, not reflecting its complete capabilities.
   - **Response**: The authors provided results for all stages of SDE-I on the full benchmark using DeepSeek V2.5 and GPT-4o-mini, demonstrating the agent's performance across different models and stages. They also noted the high costs associated with running full evaluations.

2. **Lack of Comparative Baselines (ffwo):**
   - **Concern**: No comparisons with existing agents, reducing the benchmark's context in the broader landscape.
   - **Response**: The authors evaluated the top-performing agent from the SWE-bench leaderboard (OpenHands + CodeAct v2.1 with Claude-3.5-Sonnet) on Commit0, providing comparative results. They highlighted that significant modifications were needed for OpenHands to work with Commit0, underscoring the benchmark's complexity.

3. **Insufficient Examples (ffwo):**
   - **Concern**: Lack of example instances from Commit0 and trajectories from SDE-I.
   - **Response**: The authors provided examples of Commit0 tasks and trajectories of SDE-I's performance across all stages, linking to anonymous repositories and Google Drive files. They committed to including these examples in the appendix of the final version.

4. **Cost Considerations (ffwo, xSyA):**
   - **Concern**: The high computational and financial costs of running the benchmark were not addressed.
   - **Response**: The authors added a detailed cost analysis for running SDE-I across different stages and models, including both the lite and full splits of Commit0. They contextualized costs by comparing them to historical pricing trends for LLMs.

5. **Potential Data Contamination (ffwo):**
   - **Concern**: Unclear methods for preventing agents from accessing existing implementations online.
   - **Response**: The authors explained their approach to detect and void generations using GitHub links, emphasizing that no such instances occurred in their experiments. They committed to clarifying this in the revised manuscript.

6. **Limited Insights (FDu7):**
   - **Concern**: The paper did not provide sufficiently novel insights compared to existing benchmarks.
   - **Response**: The authors outlined three actionable research directions enabled by Commit0: optimizing generation units, leveraging long-context models, and enhancing error handling techniques. They also provided preliminary experiments and findings to support these directions.

7. **SDE-I Design (FDu7):**
   - **Concern**: The choice of module-level generation might be suboptimal and could skew results.
   - **Response**: The authors clarified that modules in the lite split are relatively short, fitting within model context limits. They also provided experimental evidence showing that module-level generation outperforms function-level generation, attributing this to the holistic understanding of function interactions within a module.

8. **Fixed Budgets for Baselines (xSyA):**
   - **Concern**: Fair comparisons of different agents require fixed computational budgets.
   - **Response**: The authors proposed using cost as a proxy for computational resources, plotting average pass rates against costs. They also conducted experiments varying the number of independent generations and iterations, showing performance trade-offs.

#### Final Decision Weighting
The authors' thorough responses and manuscript revisions significantly addressed the reviewers' concerns, particularly regarding evaluation completeness, comparative baselines, and cost considerations. The addition of examples and detailed insights enhances the paper's transparency and value. While some limitations remain, such as the potential novelty of insights and the complexity of preventing data contamination, these are balanced by the benchmark's ambitious scope and potential to advance research in code generation.

---

### Decision · Program_Chairs · 2025-01-22

Accept (Poster)